

# High-frequency glacial lake mapping using time series of Sentinel-1A/1B SAR imagery: An assessment for southeastern Tibetan Plateau

Meimei Zhang[1], Fang Chen[1,2,3], Bangsen Tian[1], Dong Liang[1,2], Aqiang Yang[1]

[1]Key Laboratory of Digital Earth Science, Institute of Remote Sensing and Digital Earth, Chinese Academy of Sciences, No. 9 Dengzhuang South Road, Beijing 100094, China
[2]University of Chinese Academy of Sciences, Beijing 100049, China
[3]Hainan Key Laboratory of Earth Observation, Institute of Remote Sensing and Digital Earth, Chinese Academy of Sciences, Sanya 572029, China

*Correspondence to*: Fang Chen (chenfang_group@radi.ac.cn)

**Abstract.** Glacial lakes are important component of the cryosphere in the Tibetan Plateau. In response to climate warming, they threaten the downstream lives, ecological environment and public infrastructures through outburst floods in a short time. Although most of the efforts have been made to extract glacial lake outlines and detect their changes with remotely sensed images, the temporal frequency and spatial resolution of glacial lake datasets are generally not fine enough to reflect the

detailed process of glacial lake dynamics, especially for potentially dangerous glacial lakes with high-frequency variability. By using a full time-series Sentinel-1A/1B imagery during a year, this study presents a new systematic method to extract the glacial lake outlines with fast variability in southeastern Tibetan Plateau at the time interval of six days. Our approach was based on the level-set segmentation, combined with a median pixel compositing of SAR backscattering coefficients stacks as regularization term, to robustly estimate the lake extent across the observed time range. The mapping results were validated

against with manually digitized lake outlines derived from GF-2 PMS imagery, with the overall accuracy and Kappa coefficient of 96.54% and 0.95, respectively. In comparison with results from classical supervised SVM and unsupervised ISODATA methods, the proposed method proves to be much more robust and effective to detect glacial lakes with irregular boundaries and that have similar backscattering with surroundings. This study also demonstrates the feasibility of time-series Sentinel-1A/1B SAR data in continuous monitoring of glacial lake outline dynamics.

# 1 Introduction

Glacial lakes are important indicators of regional glacier dynamics in response to climate warming and changing precipitation (Bajracharya and Mool, 2014;Thompson et al., 2012;Capps et al., 2010). With continuous retreating and thinning of mountain glaciers in Tibet Plateau, a large number of glacial lakes have been developed and experienced rapid expansion in recent decades, leading to the increased hazard risk of glacial lake outburst floods (GLOFs) (Emmer, 2018).



These GLOFs usually evolved and erupted in a short time and can trace long distances from high altitude regions, which will pose great threat to the life and property in the downstream valleys (Veh et al., 2019;Prakash and Nagarajan, 2017).

Knowledge about the glacial lake distribution and their development are of great importance for the better understanding of the associated glacial process and hazard assessment of GLOFs (Bolch et al., 2008;Liu et al., 2014). Multitemporal mapping of glacial lakes is the first step in the evaluation of potential hazard from GLOFs, since it can be used to perform

the analysis of the evolution patterns of glacial lakes in order to identify the potentially dangerous glacial lakes on the regional scale (Song et al., 2016a;Wang et al., 2012b). Terrestrial surveying have been served as the traditional tools for studying glacial lakes for several decades. However, one of the great limitations of these techniques is that they have difficulties to access to generally remote glacial lakes over a large and risky mountainous terrain area, and they are extremely costly in terms of time and money. In contrast, remote sensing techniques can provide an archive of satellite images obtained

both from optical and microwave sensors and covering a long time, are the only feasible tools for the detailed investigation of glacial lakes without the great expense of field measurements (Zhang et al., 2015).

Optical remote sensing data, such as Landsat series satellite data, Sentinel-2A/2B, ASTER, SPOT-5 and MODIS data, have been widely applied for the studies related to the glacial lake dynamic monitoring and hazard management (Li and Sheng, 2012;Round et al., 2017;Moussavi et al., 2016;Li et al., 2018). Based on these data, visual interpretation that

depending on the subjective empirical judgment, is the main way for the delineation of mountainous glacial lakes characterized by complex terrain conditions and glaciated environments (Huggel et al., 2002;Quincey et al., 2005;Zhang et al., 2015). Many studies have usually used threshold segmentation of normalized difference water indices (NDWIs) for classifying glacial lakes (Nie et al., 2017;Song et al., 2016b;Li and Sheng, 2012). Recently, a systematic approach that integrates the advantages of the threshold segmentation method and the C-V model was proposed, which can effectively

extract glacial lakes by removing the noise from the surroundings, mountain shadows, adjacent glaciers (Zhao et al., 2018). For the accurate extraction of glacial lakes from heterogeneous backgrounds, an automated scheme that used the non-local active contour approach based on the modified NDWI produce highly reliable glacial lake maps across the entire Tibetan Plateau region (Chen et al., 2017). Although good results have been achieved through these techniques, the data acquisition are strongly hampered by weather conditions and time of day. Selection of high-quality optical images for lake mapping over

large glaciated areas is not an easy task. Besides, most of these studies have tried to monitor the glacial lakes only covering a few time intervals. Because glacial lakes that prone to outburst are likely to experience dramatic changes during several days, high-frequency mapping offer the valuable information for early identification of related hazards.

In this paper, southeastern Tibetan Plateau was chosen as the test site because glacial lakes in this region are more sensitive to climate change, abundant precipitation, and high ice-layer temperature, shown high intra-annual variability in the

lake extent. Moreover, interfered by cloudy and rainy climate, the limited data availability made the exploration of glacial lake distribution and evolution patterns even more challenging. Synthetic aperture radar (SAR) data has the advantages of all-weather and full-time capabilities, this study utilizes the free availability and short revisit cycle (six days) of Sentinel-1A/1B SAR data, and aims to develop an effective, automated and high-temporal resolution methodology for extracting the



glacial lake outlines. We combine the full time series of Sentinel-1A/1B data over southeastern Tibetan Plateau using a pixel
compositing approach of derived backscattering coefficient stacks, examine medians to robustly estimate the glacial lake
extent for each date using an improved level-set method, and demonstrate the ability of this median-based compositing
approach to deal effectively with data quality, abnormal value and other issues that commonly affect single scene
methodologies.

## 2. Study area and data

### 2.1 Study area

Our study area is located in the southeastern Tibetan Plateau, which includes parts of two sub-basins: Salween Basin and
Brahmaputra Basin (corresponding to the area of one ascending and descending Sentinel-1A/1B frames, see Figure 1). The
climate of this region is mainly influenced by two atmospheric circulation systems (Yao et al., 2012). The dry season period
is from November to April, with little precipitation and cold climate provided by the southern branch of the mid-latitude
westerlies (Wang et al., 2012a). May to October is the wet season, during this time the westerlies become weak and
meanwhile, the Indian monsoon penetrate the region, indicating the beginning of rainy season. The heavy rainfall in the wet
season accounts for 60-90% of the annual total (Wang et al., 2017). The terrain and geomorphology of southeastern Tibet are
rather complex and diverse, formed a landscape of alpine canyons. Monsoonal temperate glaciers widely developed across
the region. These monsoonal temperate glaciers are more sensitive to climate warming and abundant precipitation, and
thinning and melting faster than continental glaciers. Affected by the combined interactions from local climate conditions
and higher ice-layer temperatures, glacial lakes in this region are much larger and deeper, and generally shown navy blue in
color (Zhao et al., 2018;Song et al., 2016a).

**Figure 1. Geographic location of study area, with topographical characteristics and distribution of mountain glaciers in the two**
**river basins. Pink and blue rectangles represent frame of ascending and descending Sentinel-1 images, respectively. Green**
**rectangles indicate the coverage of GF-2 PMS images used for accuracy evaluation of the glacial lake mapping results.**

### 2.2 Data

In this study, high-frequency glacial lake extent were mapped based on the time series of C-band Sentinel-1A/1B Ground
Range Detected (GRD) images from the ascending and descending orbits at a spatial resolution of 10m. All available image
were accessed from Google Earth Engine cloud computing platform (https://code.earthengine.google.com/) with consecutive
time interval of six days, and mainly acquired during the May to October 2018, in order to ensure that most of glacial lakes
were not frozen and easily visible in the images (Nie et al., 2017). Six cloud-free images of Gaofen-2 (GF-2) panchromatic
multi-spectral (PMS) images acquired in July to August 2018 at descending orbits were used for validating mapping results
of glacial lakes. Here the GF-2 scenes were chosen due to the fact that they have high spatial resolution of 1m/4m in the
panchromatic/multi-spectral bands and a revisit cycle of five days that quite similar to the revisit period of dual satellite





constellation of Sentinel-1A and Sentinel-1B. The basically consistent acquisition dates for the Sentinel-1A/1B and GF-2 images can reduce the influence of lake extent fluctuations on the two-date lake maps comparison. DEM data were obtained to serve as the topographic reference for the geocoding of SAR and optical data, and removing the mountain shadows that may induce the classification errors for lake mapping. This study used the ASTER Global Elevation Model Version2

(GDEM V2) gridded data with a 30-m resolution collected from the website of http://gdem.ersdac.jspacesystems.or.jp/.

## 3. Methods

Figure 2 shows the detailed mapping procedure of our proposed method for glacial lakes. We first gather time-series Sentinel-1A/1B GRD images and pre-process to generate a calibrated, ortho-corrected product for water identification. Then an improved level-set segmentation method was developed for extracting glacial lake outlines in a single date based on the

median composite of backscattering coefficients. Finally, the derived Sentinel-based glacial lake outlines were validated with the manual delineation of lake boundaries from GF-2 PMS images. We describe each process in more detail in the following subsections.

**Figure 2. Flow chart of our method for mapping the glacial lakes at high temporal resolution based on time series of Sentinel-**
**1A/1B SAR data.**

### 3.1 Pre-processing of images

Each Sentinel scene was pre-processed with Sentinel-1 Toolbox using the following steps: (1) Thermal noise removal; Removes additive noise in sub-swaths to help reduce discontinuities between sub-swaths scenes (Berg et al., 2015); (2) Radiometric calibration; Computes backscatter intensity using sensor calibration parameters in the GRD metadata; (3)

Terrain correction using ASTER GDEM V2 to convert data from ground range geometry in the range-Doppler geometry. The final terrain-corrected values are converted to decibels via log scaling 10*log10(x). (4) Mountain shadow mask; Since the potential lake areas are generally on a flat surface, a slope threshold of 15° was set to exclude the mountain shadows (Zhao et al., 2018).

The GF-2 PMS imagery were also geocoded on a Universal Transverse Mercator map projection to assess the accuracy of

lake extraction results from SAR data. Because the spatial resolution of the multispectral image was not that high for the use in the precise evaluation of SAR-based lake maps, the reference lake outlines were manually delineated through visual interpretation of panchromatic image using multispectral image to aid the delineation process.

### 3.2 Extracting the time series of glacial lake outlines

For each Sentinel tile observation, the backscattering coefficient at each pixel was calculated, this creates an ensemble of
backscattering coefficient tile images. Individually, due to outliers and other artefacts, each backscattering intensity image





can be used to roughly estimate the glacial lake area using a threshold being automatically initialized based on image histogram for water and land (Figure 3). As a result, initial segmentation results for glacial lakes in each date were generated.

To effectively deal with the noise factors in a single image classification, we perform the pixel–based compositing of backscattering coefficient images within the whole observation period. The image compositing approaches have been

devoted to create a composite with pixel values that most representative of the full ensemble dataset, and unaffected by the outliers or noise in each data (Sagar et al., 2017). In this study, because we are dealing with the single-band images of backscattering coefficient values and in order to estimate the most representative value in a simple and well-understood way, we take the median value at each pixel through the ensemble of intensity images. Using the median will help to avoid a lot of potential problems with the SAR data observations, such as the presence of speckle noise that may affect individual intensity

values, residual terrain shadows not completely removed by slope measurements, and shallow water or land with high moisture content that may result in misclassification (Strozzi et al., 2012;Capps et al., 2010).

During the implementation of level-set segmentation method (Silveira and Heleno, 2009;Brox and Weickert, 2006), the median composite of backscattering coefficient was added as regularized evolution term for level-set function to avoid reinitialization and improve computational efficiency. The initial outline of glacial lake derived from threshold segmentation

served as good initial contour, as it is usually very close to the lake shape being segmented, saving more time for the contour evolution. Finally, the glacial lake outlines were progressively updated for each acquisition date. This method is fast and effective with low labor intensity due to the rough location of a given lake and the regularization term of median values.

**Figure 3. Conceptual graph of the basic steps in the process of mapping time series of glacial lakes.**

**3.3 Validating the glacial lake mapping results**

Sentinel-based glacial lake maps from July to August 2018 generated using the proposed method were assessed by comparing them against the glacial lake outlines that manually digitized from the GF-2 PMS images during the same time range. Due to the variable nature of some glacial lakes conditions and also the surrounding environment, the difference of the image acquisition date between Sentinel-1A/1B and GF-2 images were no more than five days in order to ensure the

reliability of evaluation results. Commission and omission errors were calculated beneath all lake and land pixels together. A low percentage error in the water and land classes suggests an accurate representation of lakes and surrounding mixed pixels. Moreover, Kappa coefficient and Overall accuracy were also computed from errors calculated on a number of individual glacial lakes covered by the area of GF-2 imagery. In addition to the above detailed accuracy analyses, further validation of the robustness of the proposed method was undertaken by conducting the comparisons with glacial lake maps produced by

using classical supervised SVM and unsupervised ISODATA classification methods.





## 4. Results and discussion

### 4.1 Accuracy assessment

An accuracy assessment was performed on the mapping results of some test glacial lakes. As shown in Figure 4(a), the extracted glacial lake outlines from Sentinel-1A/1B image (red contours) are almost coincident with the manually digitized

glacial lake outlines (yellow contours), these clearly illustrate that the proposed method works well for detecting glacial lakes from SAR data. To show more details about the accuracy assessment results, a relatively large glacial lake was selected as the example shown in Figure 4(b). From Figure 4(b), it can be seen that the misclassification mainly occurs along the lake outline, and the number of commission and omission pixels are low. This can be further confirmed from the enlarged distance buffer zone around the outline of this glacial lake (Figure 4(c)), where the distance between the extracted and

manually digitized glacial lake outline is less than five pixels.

**Figure 4. Extraction results for some test glacial lakes. (a) Comparison of manually digitized glacial lake outlines from GF-2 PMS images and extracted outlines from Sentinel-1A/1B images using the proposed method; (b) Commission errors (indicated as yellow color pixels) and omission errors (indicated as red color pixels) of the typical glacial lake; (c) Enlarged distance buffer zone (in**
**pixel) image around the outline for glacial lake in (b).**

To show the advantages of the proposed method, we also utilized the supervised SVM and unsupervised ISODATA methods for comparison. Figure 5 illustrates the differences between the glacial lake outlines extracted by the three methods. For the glacial lake (a) that has irregular shape and stream outlets in the lower reaches of the lake (indicated as red ellipse in

Figure 5(a)), the SVM overestimated this water area but it was completely ignored by the ISODATA, reflecting their poor ability in the delineation of glacial lakes with complex curved shapes (Tian et al., 2017). However, the proposed method takes the median composite of the backscattering coefficient values as the regularization term to restrain the lake boundary, can accurately extract the water bodies in the lower reaches region of glacial lake. For some glacial lakes that have similar backscattering with the backgrounds, as is shown in the eastern part of the glacial lake (b) (Figure 5(b)), mainly refers to the

local low areas and mud patches characterized by high moisture content, there still would be a tendency to overestimate the glacial lake area using SVM from single image. The ISODATA method extracted a rough water boundary, this is because it applies global statistics to generate clustered center, often hindered in highly complex images, especially for the complex water conditions (Tulbure and Broich, 2013;Xie et al., 2016). On the contrary, the proposed method exhibited the best performance due to its superiority in terms of spectral and backscattering variability. The last glacial lake (c) can be

accurately identified using all the three methods (Figure 5(c)), but our method outperformed the SVM and ISODATA since it can capture more local details and obtain a sharper border along the edge of the lake.

**Figure 5. Comparison of the glacial lake extent extracted from Sentinel-1A/1B data in 2018 using SVM, ISODATA and the proposed method for three sampled glacial lakes: (a), (b) and (c).**





Table 1 summarizes the accuracy statistical results for numerous glacial lakes in the GF-2 PMS images covered area. Generally, the higher omission errors relative to commission errors demonstrated that the missing glacial lake pixels are the main source of classification errors for all the three methods. The highest accuracy, with the Kappa coefficient of 0.95 and overall accuracy of 96.54%, were achieved for the classification results obtained by the proposed method. The commission and omission errors for the water and land classes are also very low, no more than 4%. The lowest Kappa coefficient and

overall accuracy are for the ISODATA method produced map (Kappa=0.79, overall accuracy=88.87%). Better results were achieved for SVM classification method, which needs a lot of user expertise and proper training samples, has a Kappa coefficient and overall accuracy of 0.85 and 92.13%, respectively. These quantitative evaluation results show that the proposed median-based composite method is effective and accurate enough to extract glacial lakes from the SAR intensity data.


**Table 1. Summary of classification accuracy of three different methods.**

### 4.2 Temporal variation of glacial lakes during the year of 2018

In the high mountainous areas, some glacial lakes develop rapidly, i.e. within some weeks or a few months (Kääb et al., 2013;Nagai et al., 2017), low-frequency monitoring is not sufficient to track changes and timely detect hazardous

developments. On these conditions, high-frequency monitoring would be highly valuable for the early detection of related hazards. Here, we examine the applicability of the developed method for the monitoring of glacial lakes with high seasonal variability in the study area. Figure 6 shows examples of multi-temporal glacial lake change detection from Sentinel-1A/1B SAR data using the proposed method. In these two dynamic glacial lakes we see different types of extent change. At the north-west extents of the first lake we see the lake water extends outwards, followed by a filled gap (indicated by red ellipse)

over the short time, highlighting the potentially unexpected nature of moraine instabilities in the region as a result of perennial water flows. In the second example, we see the complex shape at the western part of the glacial lake shifting significantly over time, with a pattern of erosion and expansion as the outflow of the lake water that almost completely detaching the lake, and then enlargement and coalescence of two small glacial lakes. The images shown in Figure 6 are only two snapshots of the continued evolution of glacial lakes through time series, and we are able to investigate the full extents

of these regions of glacial lake dynamics.

**Figure 6. Variations in the extent of glacial lakes from 3 July to 2 August.**

### 4.3 Glacial lake mapping results in the study area

In order to test the performance of the Sentinel-1A/1B data with the developed median-composite based segmentation

method over large areas, experiments covering the whole study region of southeastern Tibet Plateau were carried out, as shown in Figure 7. Three sub-regions that were characterized by different environmental conditions were selected as typical



examples for more local details. The glacial lakes have been formed in the junction area of Salween and Brahmaputra Basins, and are more densely distributed in the Brahmaputra Basin. Almost all the glacial lakes in this region can be categorized into unconnected glacial lake type, namely lakes that are not in directly contact with current glaciers, due to the accelerating

retreat of glaciers induced by warm and humid climate. Some of these glacial lakes are impounded by block or debris dams (Quincey et al., 2007), exhibiting clear boundary in the SAR intensity image. However, their confusion with terrain shadow areas (e.g. yellow ellipse in Figure 7(a)) has to be taken into account because of the very low backscattering intensity similar to that of water. In this study, terrain shadows are effectively resolved using a threshold applied on slopes derived from DEM and the meaningful median estimator for the intensity values. Other lakes including large number of small glacial lakes are

embedded in undulating landscapes or impounded by rock barriers (Mergili et al., 2013), filling with water both from abundant rainfall and glacier meltwater (Figure 7(b)-(c)). These results suggest that the combination of time series of Sentinel-1A/1B SAR imagery and median composite mapping method will provide reliable and accurate information of the temporal evolution of glacial lakes over the large and rugged alpine areas, and allow the identification of potentially dangerous glacial lakes.


**Figure 7. The spatial distribution of glacial lakes (shown as red color) extracted using the developed method over the southeastern Tibetan Plateau (indicated as pink and blue rectangles). Three zoom-in maps ((a)-(c)) in the bottom show the local details in the lake outlines overlaid on Sentinel-1A/1B median composite image.**

**4.4 Discussion**

In many glacial lake mapping methods, the lack of high-frequency mapping results is a problem for some hazardous lakes with high seasonal variations in appearances of water bodies, making it difficult for timely and reliable GLOF risk evaluation. Use of glacial lake inventory dataset with coarse temporal resolution may lead to the incorrect analysis of evolution patterns of glacial lakes which may also affect decision making in hazard management. This study devise a method that was devoted to increase the frequency maps of glacial lakes by utilizing time series of Sentinel-1A/1B SAR data with

consecutive 6-day periods. In addition to high-frequency characteristics, our method was also shown the improvement of accuracy when compares with SVM and ISODATA methods in classifying complex geometrically-shaped glacial lakes with various surroundings. It should be noted that in our study, to reduce the influence of data noise and uncertainty from single image, all the accessible images acquired during the observation period were used to compute the median value in each pixel for the precise delimitation of the lake extent. In classifying images that have not sufficient enough image number and also

high data quality, the derived median values may differ slightly and affect the extraction results of glacial lakes from what is mapped in this study.

  As for the selection of features for segmentation, in this paper, we use the SAR intensity images to extract the glacial lake outlines, which have extensive applications for water delineation due to its low backscattering values at all microwave wavelengths and distinct separation from the snow, glacier and debris cover around the glacial lakes (Strozzi et al.,

2012;Zhang et al., 2019). These allows fairly accurate classification of glacial lakes in intensity images. Given the fact that



relative instability of glacial lake water will significantly decrease the degree of interferometric synthetic aperture radar (InSAR) coherence, the capability of coherence has been recently explored to characterize and map the unconnected glacial lakes in the Himalayas (Zhang et al., 2019), proved to be a promising feature for deriving lake boundaries and may be used as input feature in this study. Furthermore, the application of the proposed method is not limited to SAR data, it can be

extended for the extraction of glacial lake outlines by incorporating spectral features from optical images, such as single-band reflectance features, water indices, and image transformation features, etc.

## 5. Conclusions

High-frequency continuous monitoring of glacial lake dynamics is crucial for the understanding of their distribution, evolution, and the driving mechanism of rapid expansions. However, we currently have limited information of glacial lake

dynamics at high frequency due to the low availability of data and lack of advanced techniques. Using time series of Sentinel-1A/1B SAR intensity data at six days interval, the new method introduced in this paper contributes to improve the accuracy of glacial lake mapping and change detection at high temporal resolution for various environmental studies and applications. This method uses a simple and systematic median composite technique of backscattering coefficient for enhancing the water separability, and meanwhile, to remove part of shadow and dark surface noises, which are often the

major causes of misclassification in the mapping of glacial lakes. The median composite values were used as the regularization term in level-set function to smooth the segmentation boundaries and prevent the occurrence of small, isolated regions in the classification of individual image. The manually digitized glacial lake outlines based on the GF-2 PMS imagery were served as reference data to validate glacial lake mapping results. Comparing with the SVM and ISODATA classification approach, the new method was shown to extract glacial lakes with highest overall accuracy of 96.54%,

particularly in mountainous areas where there are complex terrain conditions. This method also shown to be effective under large glaciated areas in southeastern Tibetan Plateau. Nevertheless, in our test cases, we did not consider the influence of different types of glacial lakes, such as proglacial lakes and supra-glacial lakes. Therefore, the robustness of the new method needs to be tested in different regions under various climatic and cryospheric backgrounds. More sites over the whole Tibet Plateau may need to be included for a thorough evaluation of the performance of the method.


**Code/Data availability.** Code/Data will be made available on request.

**Author contributions.** Meimei Zhang designed the study, developed the proposed flowchart and wrote the manuscript; Fang Chen performed the analyses; Bangsen Tian, Dong Liang collected the GF-2 PMS data for validation, Aqiang Yang

performed the SVM and ISODATA classification experiments; all the authors discussed the results.

**Competing interests.** The authors declare that they have no conflict of interest.



**Acknowledgements.** This research was supported by the National Key R&D Program of China (Grant No.2017YFE0100800), the International Partnership Program of the Chinese Academy of Sciences (Grant No.131211KYSB20170046), and the National Natural Science Foundation of China (41871345).

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

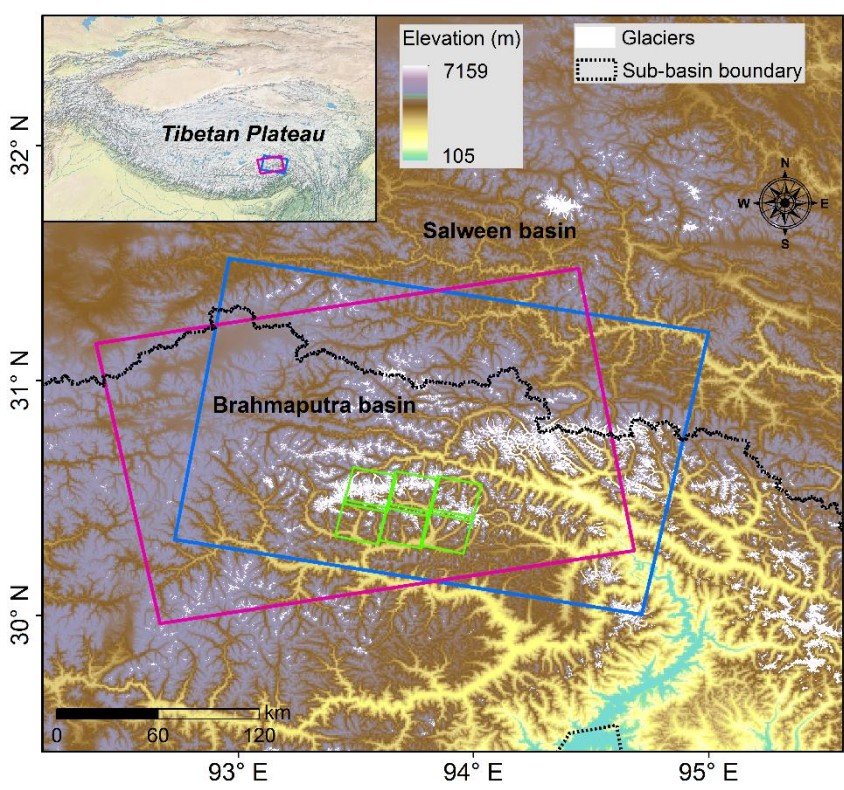

**Figure 1. Geographic location of study area, with topographical characteristics and distribution of mountain glaciers in the two river basins. Pink and blue rectangles represent frame of ascending and descending Sentinel-1 images, respectively. Green rectangles indicate the coverage of GF-2 PMS images used for accuracy evaluation of the glacial lake mapping results.**

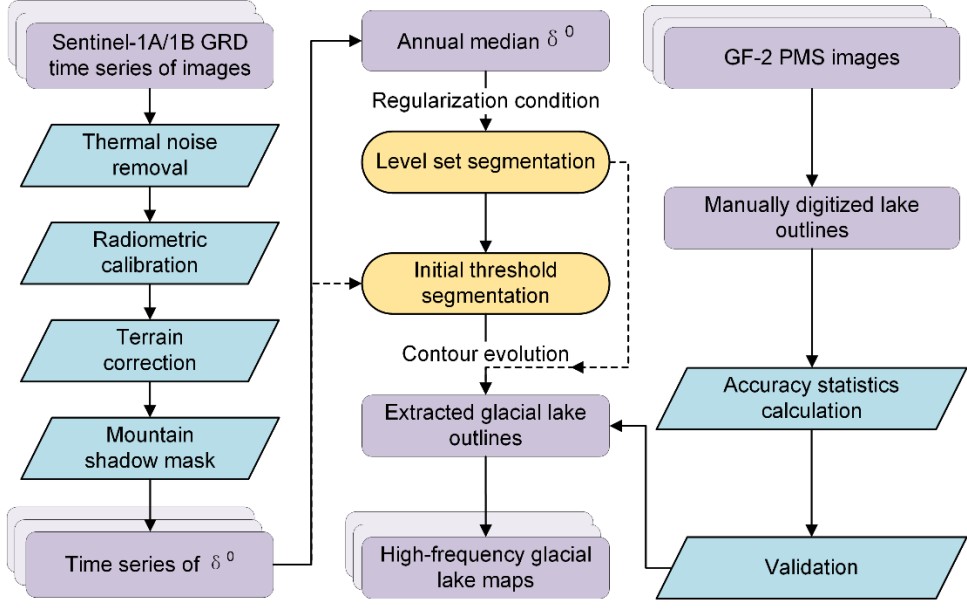




**Figure 2. Flow chart of our method for mapping the glacial lakes at high temporal resolution based on time series of Sentinel-1A/1B SAR data.**

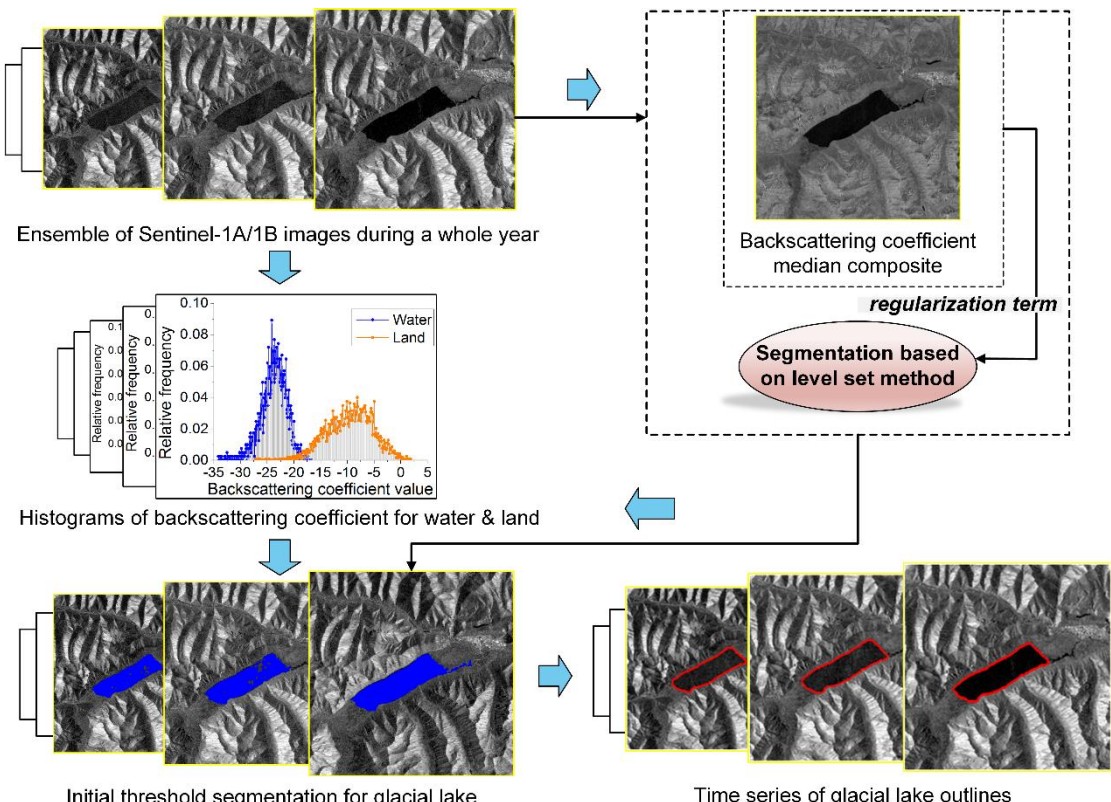


**Figure 3. Conceptual graph of the basic steps in the process of mapping time series of glacial lakes.**

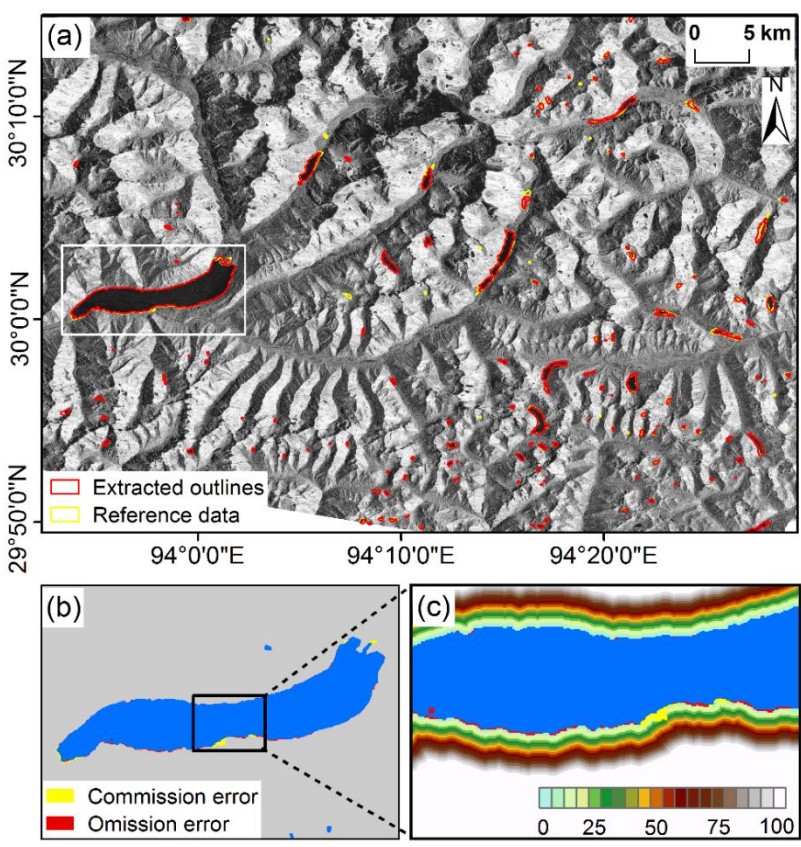

**Figure 4. Extraction results for some test glacial lakes. (a) Comparison of manually digitized glacial lake outlines from GF-2 PMS images and extracted outlines from Sentinel-1A/1B images using the proposed method; (b) Commission errors (indicated as yellow color pixels) and omission errors (indicated as red color pixels) of the typical glacial lake; (c) Enlarged distance buffer zone (in pixel) image around the outline for glacial lake in (b).**


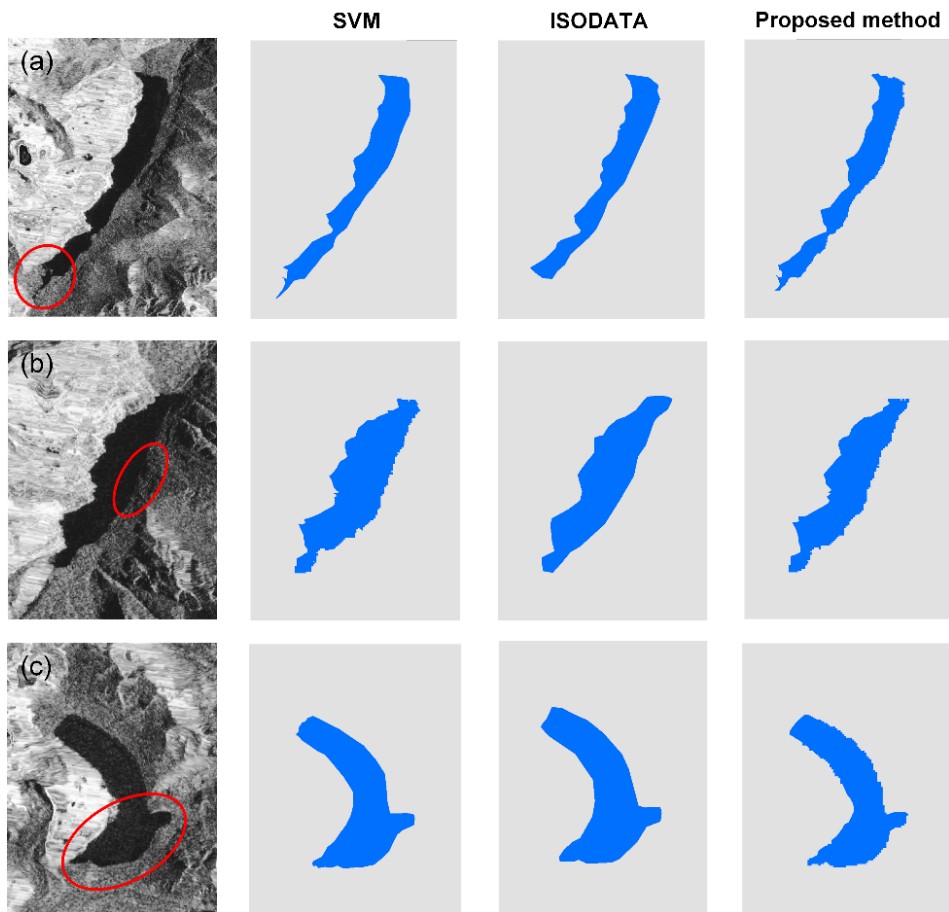

**Figure 5. Comparison of the glacial lake extent extracted from Sentinel-1A/1B data in 2018 using SVM, ISODATA and the proposed method for three sampled glacial lakes: (a), (b) and (c).**

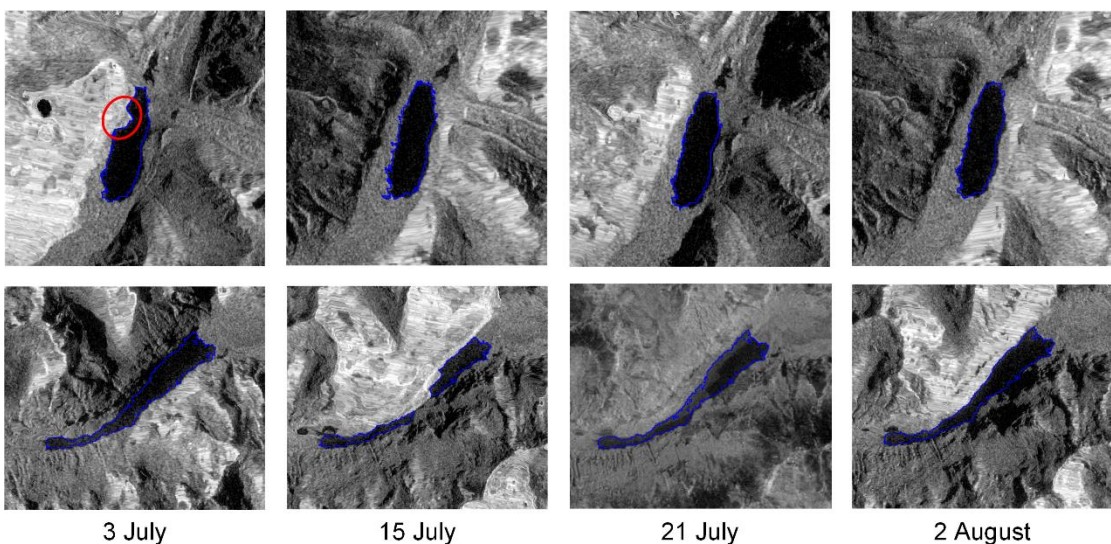


**Figure 6. Variations in the extent of glacial lakes from 3 July to 2 August.**

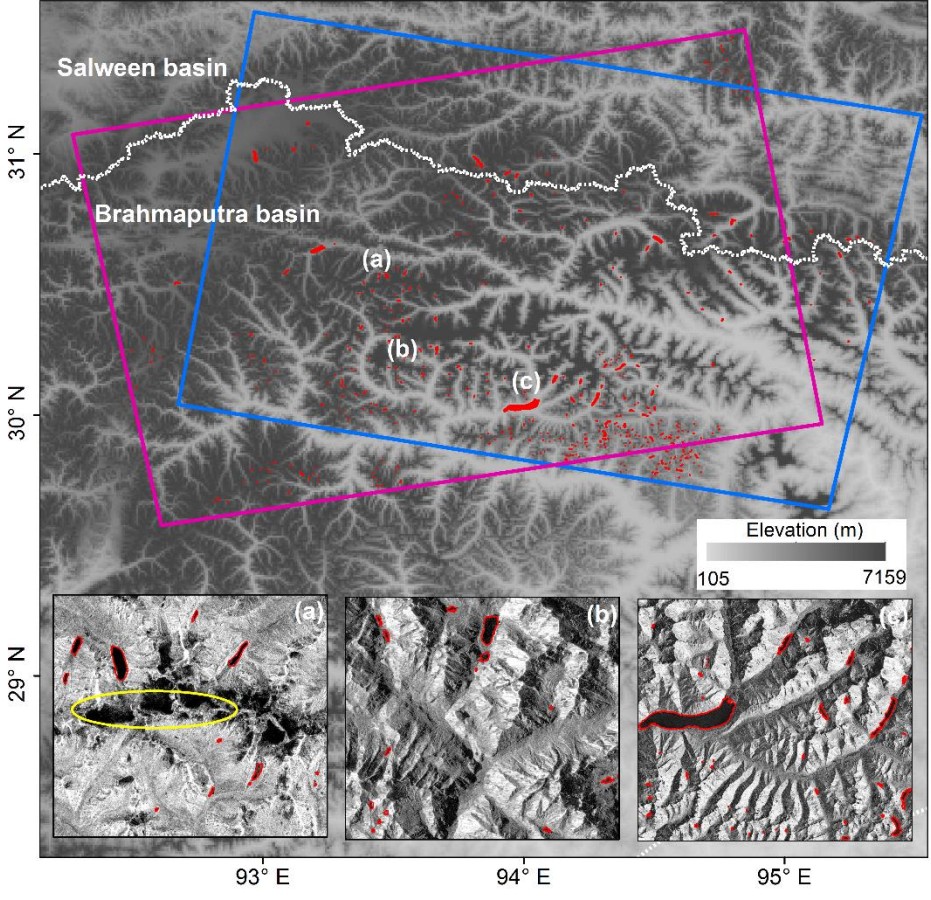



**Figure 7. The spatial distribution of glacial lakes (shown as red color) extracted using the developed method over the southeastern Tibetan Plateau (indicated as pink and blue rectangles). Three zoom-in maps ((a)-(c)) in the bottom show the local details in the lake outlines overlaid on Sentinel-1A/1B median composite image.**


**Table 1. Summary of classification accuracy of three different methods.**

| Method | Class | Commission error (%) | Omission error (%) | Kappa | Overall accuracy (%) |
|---|---|---|---|---|---|
| SVM | Water | 4.86 | 6.89 | 0.85 | 92.13 |
| | Land | 5.65 | 6.94 | | |
| ISODATA | Water | 7.62 | 8.95 | 0.79 | 88.87 |
| | Land | 8.49 | 9.14 | | |
| Proposed method | Water | 1.03 | 2.72 | 0.95 | 96.54 |
| | Land | 0.38 | 3.91 | | |