# Peer review of "High-frequency glacial lake mapping using time series of Sentinel-1A/1B SAR imagery: An assessment for southeastern Tibetan Plateau"

_Natural Hazards and Earth System Sciences, 2019_

## Referee Comment (RC1) · Adam Emmer (Referee) · 29 Jul 2019

General comment: This paper aims at introducing new method for the monitoring of glacial lakes with emphasis given on GLOF hazards studies, using Sentinel images. My major concern to this study is that it is not clear what is the aim of it, since the methodology itself has been in detail described by Zhang et al. (2019). If the aim is to apply this recently introduced methodology, I'm sorry to say that it is not addressed sufficiently in the current version of the manuscript, especially in the light of its potential in GLOF hazard studies. For that, I would expect examples of monitoring of lakes which are interested for GLOF studies (generally GLOF susceptible), i.e. lakes which

are proglacial, and time span all over the year. Instead, the authors show examples of lakes which are unconnected to glaciers (likely very old and stable) and only over a short period of time (July-August). Therefore, I see a mismatch between the story build in the introduction, pointing out the opportunity for employing in GLOF hazards studies, and actual content of the manuscript. While providing an insight into the evolution of selected lakes, the authors fail in providing more insights into the dynamics of lake level (and associated areal) changes in the whole studied region and all over the year. The main outcome, thus, seems to be the comparison with two other methods. While the performance of the proposed methodology is promising, I'm missing the evaluation of its performance in more challenging environment (i.e. proglacial lakes, analysed period all over the year) and the comparison with frequently used NDWI, and similar studies done, e.g. https://link.springer.com/article/10.1007%2Fs11769-012-0584-3. Considering the current length of the manuscript, there is certainly a space for addressing these issues.

Specific comments: L23-24: yes, but this is more the credit of Sentinel data L31: is this the case of the study area of Tibetan Plateau? L33-34: please explain how is dynamics of evolution related to GLOF hazards L50: please clearly distinguish from the results of the previous studies L56-57: this sentence needs to be supported by reference(s) L57-60: please provide reference(s) to support your statement L64-68: this is more a description what has been done; please define the main and the specific aims of the study L81: what is ice-layer temperature? L81: much larger and deeper compared to what? L138: regularized evolution term - not clear, please explain L146: why not to use all-year-long data? L154: what do you exactly mean by robustness here? L155: please refer to these methods properly L158: please describe these lakes in study area section L153: please specify how low (%) Figure 4c - I don't understand the meaning of this figure L176-178: not clear L206-207: not really addressed L209: rather use the name of the lakes L221: please specify these different environmental conditions in the study area section L245: this high frequency is exclusively related to the temporal resolution of the Sentinel data, not really to the method L249-251: this is not clear to

me L257: why did you choose to map only unconnected lakes? This does not make a lot of sense if claimed to be applicable in GLOF hazard studies L276-277: can you explain why not?

Unless these issues are addressed, I'm sorry that I can't recommend this manuscript for the publication in NHESS. I encourage the authors to include more analysis to meet my suggestions and submit the revised version of their manuscript.

---

## Referee Comment (RC2) · Norman Kerle (Referee) · 30 Jan 2020

Review of nhess-2019-219, High-frequency glacial lake mapping using time series of Sentinel- 1A/1B SAR imagery: An assessment for southeastern Tibetan Plateau

The purpose of the paper is to develop a methodology to map the extent of glacial lakes using Sentinel 1 data at high temporal resolution. In principle this is useful. However, the novelty and actual usefulness of the method is unclear. The SAR data processing method to delineate glacial lakes was already described in an earlier paper by the authors (Zhang et al., 2019), hence the current draft does not really add much at a scientific level. There have also been other papers that used SAR data to accurately

map lake extent (such as Strozzi et al., 2012), and those published methods in my view could also be adapted to a higher temporal resolution, hence I see no specific need for the presented method. I also don't see a significant error margin for lake extent mapping in those published papers, to justify a method that makes minor delineation improvements – that value of such accuracy increase for a hazard assessment is not discussed. Furthermore, the motivation of the paper is to do frequent lake extent mapping to allow better hazard assessment. However, recent work (Veh et al., 2019) has shown that while the number of glacial lakes has increased a lot in recent years, the number of failures (= GLOF) normalized by the number of lakes has actually decreased, and stands at a low level. It is thus not clear just what a hazard those lakes pose. The paper makes no distinction between stable lakes (that may show a seasonal size variation, but one that is not relevant from a hazard perspective), and lakes closely coupled with a rapidly changing glacier that may well be very dynamic and pose a serious hazard. As reader I also cannot tell what observed variation in lake size over a few days is actually significant, hence I fail to see how this helps to draw up better hazard maps. In sum, the paper to some extent repeats what was already published, and addresses a problem that is not actually clearly defined. Further comments below.

- L29 – "leading to the increased hazard risk of glacial lake outburst floods (GLOFs)" – first, this point is not clear. The study by Veh et all showed that while the number of glacial lakes has increased substantially in recent year, the actual failure rate per lake has actually been decreasing, hence it is not at all clear if the GLOF hazard has actually been growing. In addition, while a GLOF is a hazard, a process that can cause harm, it is not clear how you can claim that the actual risk is going up (risk is expected losses) – many of the lakes are located far from settlements, and a relatively local GLOF will not cause harm. - L30 – "evolved and erupted" – poor choice of words. Volcanoes erupt, glacial lakes do not - L33 – "Multitemporal mapping of glacial lakes is the first step in the evaluation of potential hazard from GLOFs" – it is not made clear how this extra knowledge is actually useful. Multi-temporal studies have been made before, yet how the higher temporal resolution of Sentinel actually aids in hazard process modelling is

not actually demonstrated. In my view glacial lakes are phenomena that show a natural seasonal dynamic that needs to be considered in the context of the dynamics of the glacier feeding it, but also the surrounding topography. Slightly better knowledge alone on the lake extent is of limited utility - L59 – "shown high intra-annual variability in the lake extent." No evidence for this is presented, but a clear characterization of that dynamic is needed for a reader to judge what monitoring frequency is adequate - L96 - "1B. The basically consistent acquisition dates for the Sentinel-1A/1B and GF-2" – what does that mean? No acquisition dates are reported, and given that nothing is said about the actual dynamics of the lake extent it is impossible to judge whether the GF-2 images truly coincide with the lake extents shown in the SAR data. Only later this is clarified (L149) - L205 – "low-frequency monitoring is not sufficient to track changes and timely detect hazardous developments" – in my view this is an unsubstantiated claim. I am not aware of studies that actually studied in detail the dynamics of specific glacial lakes in light of changes in hazard. Hence here we have no indication as to what frequency is sufficient to capture critical developments, whilst limiting unnecessary data processing because of an unnecessarily high monitoring frequency - L240 – "In many glacial lake mapping methods, the lack of high-frequency mapping results is a problem for some hazardous lakes with high seasonal variations in appearances of water bodies, making it difficult for timely and reliable GLOF risk evaluation" – same as above. The literature review in the manuscript has not identified such a critical gap. - L264 – "However, we currently have limited information of glacial lake dynamics at high frequency due to the low availability of data and lack of advanced techniques." A number of studies has shown the ability to use satellite data, including SAR, to accurately map glacial lake extent (e.g., Strozzi et al.). Hence I don't agree that we don't have a suitable technique – as far as I can tell the earlier developed methods can also be applied at greater temporal resolution - English is poor in places, e.g. "Terrestrial surveying have", "pixel values that most representative", "This study devise a"

2019-219, 2019.